# Assessing the Environmental Quality Resulting from Damages to Human-Nature Interactions Caused by Population Increase: A Systems Thinking Approach

**Fernando Ramos-Quintana [1],\*, Héctor Sotelo-Nava [1], Hugo Saldarriaga-Noreña [2]** and **Efraín Tovar-Sánchez [3]**

1   Dirección General de Desarrollo Sustentable, Universidad Autónoma del Estado de Morelos, Av. Universidad 1001, Cuernavaca, Morelos 62209, Mexico; hector.sotelo@uaem.mx
2   Centro de Investigaciones Químicas, Instituto de Ciencias Básicas y Aplicadas, Universidad Autónoma del Estado de Morelos, Av. Universidad 1001, Cuernavaca, Morelos 62209, Mexico; hsaldarriaga@uaem.mx
3   Centro de Investigación en Biodiversidad y Conservación, Universidad Autónoma del Estado de Morelos, Av. Universidad 1001, Cuernavaca, Morelos 62209, Mexico; efrain_tovar@uaem.mx
\*   Correspondence: ramosfernando747@gmail.com; Tel.: +52-777-352-0936

**Abstract:** Multiple interactions between population increase-as driving force- and pressure factors can cause damage to human-nature interactions. In this paper, we aim to identify, understand, and assess those interactions that exert effects on environment quality. The assessments of multiple interactions will allow selecting management actions to reduce negative effects, such as the loss of vegetation cover, on the environment. However, multiple interactions hinder the understanding of such complex systems. The relevance of this study is related to the support of the systems thinking approach to achieve two objectives: (1) to build a conceptual framework that facilitates the construction of a network aimed at representing the multiple interactions; (2) to build a closed system for the sake of developing a sustainable environmental management system. Thus, the performance of the implemented management actions is assessed through the feedback loop of the closed system. The proposed conceptual framework and the closed system were applied to the state of Morelos, Mexico. We highlight the following results: the systems thinking approach facilitated the construction of a conceptual framework to build understandable causal network; a set of environmental pathways were derived from the causal network and then combined to define and assess a global environmental state. Environmental pathways are composed of relationships between population increase and pressure variables that exert effects on the environment quality; the feedback loop facilitated the performance analysis of implemented management actions related to natural protected areas. The current results suggest further research to apply this study to diverse systems where multiple interactions between drivers and pressure factors damage human-nature interactions, thus exerting effects on the environmental state.

**Keywords:** systems thinking; closed systems; feedback; multiple interactions; networks; environmental systems

## 1. Introduction

### 1.1. The Problem of Modeling Multiple Human-Nature Interactions

The interactions of people with natural components take place within integrated systems defined by coupled human and natural systems [1]. It has been pointed out that the complexity associated with such coupled systems has not been well understood. This is mainly due to the traditional

separation of ecological and social sciences. The analysis of human-nature interactions is carried out through the theories of complexity and by considering that natural and social systems are dynamic and non-linear [2]. The definition of complexity was addressed in two ways by Ratter [2]: on the one hand, it is based on the number of elements of a system. That is, as the number of elements increases the system becomes more complex; on the other hand it is focused on the behavior of the system by taking into account the relationships and processes that occur between its elements. Thus, the behavior of the system needs to be explained through the properties of the single elements and moreover through the interactions between them. The behavioral complexity of a system is not necessarily associated with a large number of elements and relationships between them. Thus, dynamic systems with few elements but with non-linear interactions could be considered as behavioral complex systems.

Important benefits have been gained from the interactions of humans with nature through time. However, the causes and consequences of the loss of interaction of humans with nature, called the "extinction of experience", were analyzed by Soga and Gaston [3]. They pointed out that among the causes of the separation of humans from nature is the rapid growth of people living in urban areas. Likewise, the time spent on the use of electronic devices, among others. It means that outdoor nature experiences are being replaced, which can be interpreted as a kind of "extinction of experience" as termed by Pyle [4]. A model to explain the link between causes, consequences, extinction experiences, and feedback loops was proposed by Soga and Gaston [3]. Two causes related to the loss of human-nature interactions are exposed in this model: (1) the loss of opportunity (represented by the decline in opportunities to directly experience nature); (2) and the loss of orientation (the reduced emotional affinity of people with nature). Meanwhile, the consequences are represented by changes in health and well-being, as well as emotional, attitudinal, and behavioral changes. In this model, the extinction of experience leads to a feedback loop through which the interactions taking place between consequences accelerate the loss of interactions with nature. Due to this fact, the loss of opportunities and the loss of orientation increase further. The feedback loop in this case induces negative effects on human-nature interactions. However, the integration of social and biophysical systems could be enhanced by including feedback mechanisms, as pointed out by Verburg [5]. He affirmed that for the case of land use models. However, it can be certainly applied to other domains where feedback loops play an important role to support sustainable human-nature interactions. There is a tendency to believe that the benefits of contemporary societies depend on exploiting the natural world. Based on this argument, we may consider that the progress of civilization entails the subjugation of nature. Conversely, there is also evidence of nonhuman environments that enhance human physical and mental productivity and satisfaction [6].

The assessments of interactions guide the responses of human society. Such responses are related to implemented management actions to improve the state of the ecosystems and the quality of the environment. The environmental state can be composed of several key environmental variables such as $CO_2$ emissions, Waste, Water, Loss of Biodiversity, and Air Quality. These key environmental variables were considered by an OECD-Outlook to 2030 [7]. The main objective of the assessments is to provide policymakers with valuable data to facilitate the selection of adequate management actions.

The work presented in this paper is mainly focused on the damage caused by human factors to nature. These factors are associated with the state of the environment in a specific region of Mexico. Essentially, the set of multiple interactions cause complex dynamic behaviors that should be understandable to be analyzed and assessed, in the best way possible.

The multiple interactions should be organized as a set of links between individual components that belong to different categories. These can be drivers, pressure factors, impacts, implementable management actions, feedback elements, among others. Thus, the link of these categories determines a structure that should be correctly built, which can be used in the analysis and assessment processes conducted by decision makers.

### 1.2. The Population Increase and Its Effects on Key Environmental Variables

It has been documented in the literature that human activities derived from population growth represents a driving force that affects nature. The increase in human population along with social, economic and political factors determine interactions with nature that can cause direct or indirect effects on ecosystems [8]. Table 1 shows examples of effects of the population increase on key environmental variables.

**Table 1.** Some examples of effects caused by population increase on key environmental variables.

| Relationships | Observations and Statistical Data |
| --- | --- |
| Population increase causes water availability to decrease. | In 2000, 150 million people lived in cities with perennial water shortage (i.e., annual water availability <100 L/person/day) within their urban extent. Based on a forecast, in 2050, 993 million of people will live in cities with perennial water shortage within their urban extent [9]. Other related work is found in [10–12]. |
| Population increase causes increase in solid waste. | Based on a study of the population increase versus the increase of generation of solid waste in San Antonio Texas, USA: 1980: population = 786,023; solid waste = 154,983 (tons) 2010: population = 1,323,956; solid waste = 366,125 (tons) Conclusion: the population increased 168.43 % in 2010 with respect to 1980; whereas, the solid waste generation increased 236.23% in 2010 with respect to 1980. Thus, the percentage increase of solid waste generation increased was bigger than the percentage increase of population [13]. Urban waste production has risen tenfold compared with the growth of urban population [14]. Other related work is found in [15–18]. |
| Population increase brings about increases in $CO_2$ emissions. | A model for Beijing, China shows that the intensified urban development and the expanded population demand more energy consumption, thus increasing the $CO_2$ emissions. The $CO_2$ emissions will increase from 118.41 $MtCO_2$ in 2005 to 169 $MtCO_2$ in 2030 [19]. Other related work [20–23]. |
| Population increase results in increase in the transportation sector, which in turn causes $CO_2$ emissions to increase. | Population size increased 7.92%: from $1267.43 \times 10^6$ in the year 2000 to $1367.82 \times 10^4$ in 2014. GDP increased 272.50%: from $1,311,691 \times 10^6$ (euros) in the year 2000 to $4,886,144 \times 10^6$ (euros) in the year 2014. $CO_2$ emissions of the transportation sector increased 218.23%: from $32,584.03 \times 10^4$ (tons) in 2000 to $103,695.38 \times 10^4$ (tons) in 2014. The combination of population increase with the economic power increase brought about an increase of 218.23% in $CO_2$ emissions from 2000 to 2014 [24]. One more related work: [25]. |
| Population growth influences fires, thus causing losses of vegetation cover. | This research uses two concepts of wildland-urban interfaces (WUI) to study the human influence on the fire regimes of California: the interface WUI, where development abuts wildland vegetation, and the intermix WUI, where development intermingles with wildland vegetation. Based on nonlinear anthropogenic relationships the following thresholds were estimated suggesting that fire frequency is likely to be highest when population density is between 35 and 45 people/$Km^2$, the proportion of intermix WUI is ~20–30%; the proportion of intermix WUI is ~25–35%, the mean distance to intermix WUI is <9 km, and the mean distance to interface WUI is <14 km. The majority of fires are burning closer to developed areas, whose human presence is normal. Therefore, future conditions that include continued growth of intermix WUI may also contribute to greater fire risk and devastating ecological impacts may take place if development continues to grow farther into wildland [26]. Other related work [27,28]. |

In addition, roads contribute to the stress of ecology systems (a nature factor) through the modification of animal behavior, alteration of the physical environment, changes of the populations of dynamic composition of species, and the loss of vegetation cover, among others [29]; It is important to point out that the relationships between the population increase and the increase of key environmental

variables, such as $CO_2$ emissions or solid waste, among others, are not necessarily linear. It is possible that some key environmental variables increase faster than the population increase, when certain thresholds have been overcome [30].

*1.3. Conceptual Framework to Build Cause-Effect Relationships between Variables*

To facilitate the modeling of cause-effect interactions taking place in environmental problems, frameworks representing unidirectional cause-effect relationships have been proposed [31]. Under the PSR (PSR stands for Pressure, State, and Response categories) framework, *Pressure* variables exert effects on the environmental *State*, thus requiring the societal *Response* to improve the trends of environmental key variables. Unidirectional causal chains are built within a PSR framework. However, models based on unidirectional causal chains match weakly with real situations. Non-unidirectional multiple cause-effect interactions take place in real situations. Therefore, causal networks instead of causal chains are required in real situations. An extended version of the PSR framework was proposed in [32] within which the categories of Drivers and Impacts were included. This framework is known as DPSIR (Driving-Force, Pressure, State, Impacts, and Responses).

The DPSIR framework has been applied to environmental problems and other domains [33–37]. In most of the DPSIR applications one key environmental variable has been considered, such as water [38], solid waste [39], air pollution [40]. However, an environmental assessment method based on the DPSIR framework that combines the three major environmental factors water, atmosphere, and soil was proposed by Wang [41].

An important challenge is to adopt the DPSIR framework to build well-structured networks capable of representing human-nature interactions between several key environmental variables. The network will allow us to extract environmental pathways linking drivers with pressure factors. Each pathway will be associated with a key environmental variable. The pathways can facilitate the analyses and assessments of the effects of each environmental variable on a global environmental state. Therefore, pathways converge into the node that represents the global environmental state. As a result, a global environmental state index can be built by aggregating the key environmental variables.

*1.4. The Systems Thinking Approach as Support to Build Multiple Interactions*

The capacity of building adequate structures of systems submitted to multiple interactions between their components can be found in the concept known as Systems Thinking Approach. The multiple interactions between multiple factors bring about dynamic and complex systems. Systems thinking was defined as a framework for seeing interrelationships rather than individual things by Senge [42]. In addition, such framework allows seeing patterns of change rather than static snapshots that can be expressed by the reductionist models. However, they are unable to fully depict or help us to understand the complex and dynamic scenarios [43].

"Systems thinking is a set of synergistic analytic skills used to improve the capability of identifying and understanding systems, predicting their behaviors, and devising modifications to them in order to produce the desired effects. These skills work together as a system" [44].

The development of an underlying structure is the basis for making reliable inferences about the systems under study as affirmed by Richmond [45]. In addition, the behavior of such complexes systems is controlled by its dynamic structure, thus highlighting the importance of building adequate structures. Systems thinking consists of three components: elements (the characteristics), interconnections (the way these characteristics relate to and/or feedback into each other), and a function or purpose, as stated by Meadows [46]. Thus, the involved elements should be characterized for the sake of the construction of an adequate structure. The understanding of the dynamic behaviors associated with the system under study will be facilitated through the use of such structure. The ability of systems thinking to represent and assess dynamic complexity was underlined by Sweeney and Sterman [47]. "*If systems thinking leads to a deeper understanding through system dynamics, then the result*

*will be positive*", as asserted by Forrester [48]. The ability to understand interrelationships is highlighted in an important number of definitions related to systems thinking such as the case of [49–52].

The development of conceptual frameworks supported by systems thinking approaches facilitates the modeling of causal networks. The conceptual frameworks should provide, in turn, the support to link the subsystems that compose the whole system. These subsystems can be seen as part of processes with inputs and outputs associated with them. Then, a classification of the different variables and factors is made to distribute them into their corresponding subsystem. Once the process composed of subsystems has been modeled, we establish the different relationships between variables and/or factors belonging to different subsystems. The interconnections between relationships result in the causal network we want to build.

However, the problem of modeling complex systems becomes harder when the performance of the implemented management actions is assessed. Once again, the systems thinking approach helps us to address this problem through the feedback loop.

The systems thinking approach has been applied to study topics related to the environmental issues such as climate change [53], and sustainability [54–56]. Ecological, economic and social factors are included in the sustainability concept in which the interrelation between cultural, health and political aspects are involved, thus forming complex systems [57]. Therefore, Systems Thinking can help people understanding the complexity of the sustainability concept. The importance of integrating systems dynamics modeling to reveal complex interconnections, dependencies and causal relationships between sustainability indicators was denoted by Onat [58].

### 1.5. The Proposal Based on a Systems Thinking Approach

We applied the concept of systems thinking approach to study multiple interactions between drivers and pressure factors that exerts effects on human-nature interactions. In particular, we aim at studying how the effects on human-nature interactions affect the environmental state quality in Morelos, Mexico. We show how systems thinking supports the construction of a structure that helps to improve the understanding of the dynamic behavior caused by human-nature interactions. Direct and indirect interactions between human actions with nature have been considered. Direct interactions take place when human actions damage directly part of nature, then resulting in damages to the environment. Meanwhile, indirect interactions result when human actions bring about effects on variables damaging nature and the environmental state. For example, constructing roads is good for the economy but bad for ecosystems, as they destroy extensive vegetation cover.

The interactions are quantified by trends of cause-effect relationships between the involved key environmental variables. The assessment of a global environmental state results from the combination of trends of key environmental variables. Hence, the global environmental state is represented by an index, which is also called an aggregated indicator. The assessment of risky trends serves as a guide to proposing environmental management actions aimed at improving such trends over time.

## 2. Materials and Methods

### 2.1. Materials

The material required in this research is related to the data of the driving-force variable and the pressure variables. The only driving-force variable is the population increase. Meanwhile the pressure variables are: $CO_2$ emissions, Solid Waste, Water Availability, Loss of Vegetation Cover, Air Quality, Transport Vehicles, Construction of Transport Roads, and Forest Fires. The data of the involved variables in this study were compiled from different official sources for the period 2000–2010. Table 2 shows the average data per year related to the variables mentioned before.

**Table 2.** Real data of the involved variable for the period of time 2000–2010. Data compiled from different official institutions in México and USA [59–67].

| Year | Pop (Persons) (Driver-Var) | $CO_2$ (Gg) (Press-Var) | Trans-Ro (Km) (Press-Var) | FF (Ha) (Press-Var) | LVC (Ha) (Press-Var) | Wat-Av ($m^3$/per) (Press-Var) | Trans-Ve (Num. Vehicles) (Press-Var) | Was (Tons) (Press-Var) | Air-Quality ($PM_{2.5}$) (mass/$m^3$) (Press-Var) |
|---|---|---|---|---|---|---|---|---|---|
| 2000 | 1,555,296 | 2816.2 | 2001 | 12 | 90.4 | 2.818 | 155,600 | 459,000 | $1.01603 \times 10^{-8}$ |
| 2001 | 1,564,627 | 2865.27 | 2029 | 27 | 201.5 | 2.818 | 175,000 | 472,000 | No-Data |
| 2002 | 1,574.015 | 2974.88 | 2029 | 69 | 257.0 | 2.818 | 187,500 | 483,000 | $1.00925 \times 10^{-8}$ |
| 2003 | 1,583,459 | 3064.54 | 2029 | 69 | 329.7 | 2.713 | 192,500 | 493,000 | $1.11715 \times 10^{-8}$ |
| 2004 | 1,592,960 | 3231.57 | 2058 | 69 | 405.3 | 2.701 | 200,000 | 526,000 | $1.07855 \times 10^{-8}$ |
| 2005 | 1,612,899 | 3358.76 | 2080 | 69 | 476.1 | 2.746 | 212,500 | 538,000 | $1.13786 \times 10^{-8}$ |
| 2006 | 1,645,157 | 3530.68 | 2080 | 69 | 551.3 | 2.029 | 250,000 | 548,000 | $1.18449 \times 10^{-8}$ |
| 2007 | 1,678,060 | 4552.01 | 2112 | 72 | 613.7 | 2.055 | 270,000 | 551,000 | $1.28523 \times 10^{-8}$ |
| 2008 | 1,711,621 | 3652.88 | 2477 | 75.5 | 681.8 | 2.049 | 290,000 | 555,000 | $1.18733 \times 10^{-8}$ |
| 2009 | 1,745,854 | 3784.18 | 2477 | 77.5 | 762.7 | 2.040 | 310,000 | 558,000 | $1.04967 \times 10^{-8}$ |
| 2010 | 1,777,227 | 3859.22 | 2986 | 78.5 | 843.3 | 1.987 | 340,000 | 596,000 | $1.06340 \times 10^{-8}$ |

The meaning of the acronyms of Table 2 are: Pop = Population; $CO_2$ = $CO_2$ emissions; Trans-Ro = Transport Routes; FF = Forest Fires; LVC = Loss of Vegetation Cover; Wat-Av = Water Availability; Trans-Ve = Transport Vehicles; Was = Waste; Air Quality = Air Quality.

Another type of material needed for this work was provided by experts in the field of environmental issues. This material is represented by a list of implementable environmental management actions aimed at improving the state of environmental quality of a region. The environmental management actions concern the following key environmental variables: $CO_2$ emissions, solid waste, water availability, loss of vegetation cover, and air quality.

The environmental management actions have been chosen for the case of the state of Morelos, Mexico. The environmental management actions are associated with the key environmental variables calculated for the period 2000–2010. Both the key environmental variables and the environmental management actions have been chosen in accordance with the guidelines of an Outlook to 2030 made by the OECD [7]. This outlook describes a set of environmental scenarios for the year 2030. Such scenarios could result depending on the performance of the implemented management actions. The scenarios have been labeled with green light (good scenarios), yellow light (more or less good scenarios), and red light (bad scenarios). The types of implemented management actions and the way they are implemented will determine what scenarios would be reached. Table 3 shows this list of implementable environmental management actions for the case of the state of Morelos, Mexico, which were provided by environmental experts.

**Table 3.** Environmental management actions to be chosen to reduce the current environmental trends.

| Key Environmental Variables | Environmental Management Actions |
|---|---|
| Waste | • Construction of infrastructure for the separation, recycling, collection and disposal of wastes<br>• Construction of regional composting plants in areas of high organic waste generation and strategic areas for agriculture<br>• A formal inter-state program for the prevention and integral management of wastes<br>• An ongoing awareness campaign for the reduction of the generation of solid wastes |
| Water | • Modern infrastructure for an efficient management and monitoring of the continuous operation of the existing waste-water treatment plants<br>• Modern hydraulic infrastructure that ensures the extraction, the supply and adequate use of the liquid for domestic purposes<br>• The reuse of treated water to reduce the consumption of water of first quality<br>• A program of capture and use of rainwater in priority areas |
| Air-Quality | • Vehicle transport control<br>• Forest fires control<br>• Environmental education<br>• Clean production<br>• Avoiding burning the residues of the sugarcane crop by using them for fertilizer, biodigesters, and power generation, among others |
| $CO_2$ Emissions | • A program of road re-engineering along with an interstate vehicle verification with mobility restrictions, mainly within metropolitan zones<br>• Modernization of the vehicle fleet<br>• Hybrid and electric vehicles<br>• The use of alternative fuels such as ethanol and biodiesel<br>• The reorganization of loading and passenger transportation |
| Loss of Vegetation Cover | • Natural protected areas<br>• Ecological zoning of the territory<br>• Monitoring and control of forest fires<br>• Reforestation<br>• Payment of environmental services |

*2.2. Methods*

One of the most important roles of modeling is to help users understand complex systems composed of multifactorial interactions, thus paving the way for building reliable assessments.

The models and methods developed in this section depend integrally on the methodology to be developed. The proposed methodology should be based on a systems thinking approach to meet several requirements to be a useful tool for future applications. (1) It should allow building a conceptual framework capable of facilitating the understanding and assessments of multifactorial interactions between drivers and pressure variables that affect the human-nature interactions. Consequently, the effects on human-nature interactions could cause damage to the environmental state quality of a given region. (2) It should provide decision-makers with environmental assessments to support the selection of environmental management actions. (3) It should provide guidance to build a closed system able to handle feedback loops. The closed system favors the development of a sustainable environmental management system.

It is important to mention that this methodology has not been previously published, but is proposed in this paper. As most of the methodologies of this nature, it should be applied to real cases, as an important part of it. Thus, this methodology is applied to the case of the state of Morelos, Mexico using real data of the period 2000–2010. Hence, in this work we propose a methodology to build a conceptual framework to be applied to the real case mentioned before.

Following the methodology step by step helps to situate in their correct context both models and methods to be developed. This methodology uses a top-down mode going from a high level of abstraction to specific details. It is composed of the steps described below.

**Step 1.** *Describing the multiple factors related to human activities and nature*. The multiple interactions taking place between multiple factors related to diverse human activities and nature can damage the environmental state.

Figure 1 below shows a model at high level of abstraction that depicts diverse factors related to human activities that interact with diverse aspects related to nature. Most of the factors shown in Figure 1 concern the case of the state of Morelos, Mexico. Hence, in this study we work with available information using actual data from the period 2000–2010 in variables from Morelos, Mexico. Consequently, some of the factors that are included in Figure 1 are not used here because there is not available information to be considered in this study. Such is the case of the environmental education, industrial development, flooding, and droughts, to mention the most important.

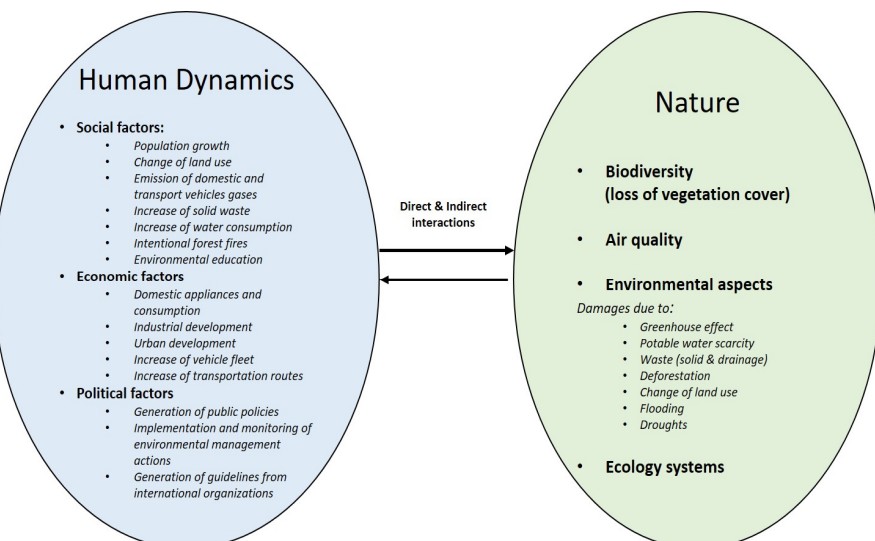

**Figure 1.** Human activities composed of social, economic, and political factors interacting with diverse aspects associated with nature.

**Step 2**. *Building the Closed System.* Figure 2 shows a model of the closed system built at high level of abstraction. It is composed of an open system and a feedback loop. Inputs and outputs for both the open system and the feedback module are described. Nevertheless, the open system needs a sequence of tasks to be achieved from the input to the output. On the one hand, the input is represented by a set of drivers and pressure factors. On the other hand, the output will provide a set of environmental management actions to be implemented. These tasks should be accomplished by a sequence of modules. Each one of these modules will develop models and/or methods to achieve their tasks.

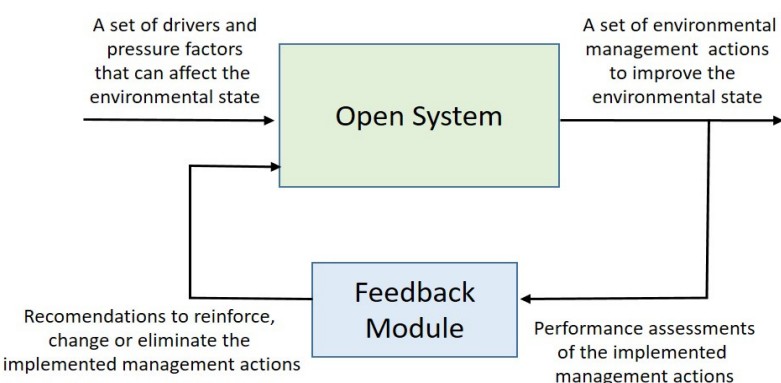

**Figure 2.** The closed system at high level of abstraction.

**Step 3.** *Building the Open System.* Three modules are required for this open system. Figure 3 shows the sequence of modules at a lower level of abstraction by describing with more details the tasks to be achieved within each module to obtain the expected outputs. As we can see in Figure 3, the function of Module 1 is to build a causal network, which is the input of Module 2. The function of Module 2 is to derive a set of environmental pathways from the causal network to produce assessments of the environmental state and the impacts on it. Finally, the function of Module 3 is to select a set of environmental management actions aimed at improving the environmental state.

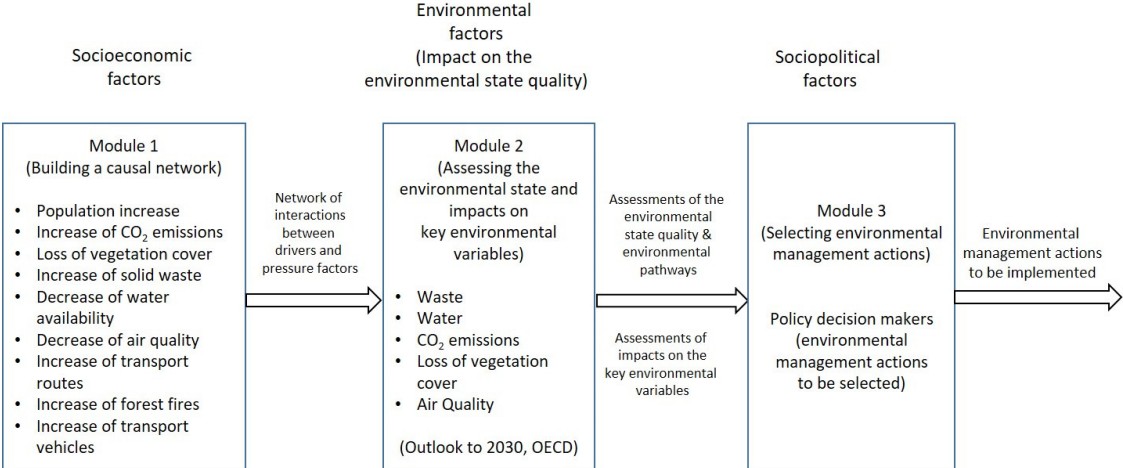

**Figure 3.** An open system that describes multiple interactions between drivers and pressure factors that affect the environmental state. At the output, a set of environmental management actions to improve the environmental quality.

**Step 4.** *Building the casual network within Module 1.* The factors shown in Module 1 of the open system described in Figure 3 are those corresponding to the study of the state of Morelos, México.

These factors are represented by drivers and pressures variables, which should be linked to build a causal network. Two methods are developed within Module 1. The first method aims to build casual relationships between drivers and pressure variables; the second method uses causal relationships to build the causal network, which will be used to assess the environmental state in Module 2. The type of causal networks built in this work is known as directed graphs, whose links are represented by arrows. The sense of the arrows is very important from the functional point of view of the network.

**Step 4.1.** *The method to build causal relationships*. Henceforward, we will represent any causal relationship by the following notation (A $\rightarrow$ B), meaning that A causes B. Most of the interactions will be represented in this work by causal relationships by taking into account the following properties: (1) the reflexive property is not considered, which means that any variable "A" cannot be related to itself; (2) the relationships are asymmetric, in other words, if A $\rightarrow$ B, not necessarily B $\rightarrow$ A, at the same proportion; (3) the transitivity property is considered by meeting the following rule: If A is related to B, and B is related to C, then A is related to C by transitivity. In general, in a sequence of causal relationships, the variable H (the head of the sequence) can be related to the variable T (the tail of the sequence) through a set of intermediate relationships, as denoted by the following expression:

$$(H \rightarrow B) \wedge (B \rightarrow C) \wedge (C \rightarrow D) \ldots \ldots (R \rightarrow S) \wedge (S \rightarrow T),$$

where "$\wedge$" represents the conjunction operator or the operator "AND". We can verify that as the number of intermediate relationships increases, the level of the relationship between H and T tends to reduce considerably.

A causal relationship (A $\rightarrow$ B) is quantified using the linear regression method, where an independent variable A causes changes on the dependent variable B. The correlation factor is used commonly in the simple regression analysis to denote whether the dependent variable responds to changes in the independent variable. However, we aim at quantifying the trend of causal relationships. Thus, the trend will be represented by the slope of the interpolated straight line, derived from the linear regression method. The interpolated straight line is represented by the expression: $Y = B_o + B_1 X$, where coefficient $B_o$ represents the value of Y when X = 0, and coefficient $B_1$ represents the tangent value of the slope of the interpolated straight line. Therefore, $B_1$ does not have units such as kilograms, kilometers, etc. The value of the slope can quantify the relationship trend which could be either in an upward or a downward direction. This statement is derived from the following reasoning: if the slope value of $B_1$ tends to increase, then the trend of the relationship points towards an upward direction; on the contrary, if the slope value of $B_1$ tends to decrease, then the trend of the relationship points towards a downward direction. We recall that the tangent values can vary from 0 to $\infty$ (for positive values) or between 0 and $-\infty$ (for negative values). However, the tangent values can be transformed into angular values using the following function: $\text{angtan}(\alpha) = \theta$ or $\text{tg}^{-1}(\alpha) = \theta$, which can be read as follows: $\theta$ is the angle whose tangent is $\alpha$. Hence, instead of using values between 0 and $\infty$ or between 0 and $-\infty$, we can use values between $0°$ and $90°$ or between $0°$ and $-90°$, which is easier to be interpreted. Likewise, the values of the ranges $[0°, 90°]$ or $[0°, -90°]$ can be converted into normalized values between 0 and 1 or 0 and $-1$, as follows: Normalized-value = $\theta/90$, where $\theta$ represents the current angular value of the slope to be normalized.

The variables to be related are shown in Table 1 described in Section 2. Two categories are involved in Table 1: drivers and pressure variables. In this work, we have only one driver, which is the population increase. Meanwhile, the remaining variables of Table 1 belong to the category of pressure variables. We aim to build a causal network representing cause-effect relationships between the population (the driving force variable) and the pressure variables and more precisely with the key environmental variables. Table 4 shows the cause-effect relationships between the population increase (the driving force variable) and pressure variables.

**Table 4.** The cause-effect relationships between the population increase (the driving-force variable) and pressure variables.

| Causal Relationships | Type of Relationship | Observations/Interpretations |
|---|---|---|
| $\Delta$Pop $\rightarrow$ $\Delta$Was | Direct | *If* population increases, *then* waste generation increases |
| $\Delta$Pop $\rightarrow$ -$\Delta$Wat-Av | Direct | *If* population increases, *then* water availability decreases |
| $\Delta$Pop $\rightarrow$ $\Delta$Air-Quality | Direct | *If* population increases, *then* the air-quality decreases |
| $\Delta$Pop $\rightarrow$ $\Delta$Trans-Ro | Direct | *If* population increases, *then* transportation roads increase |
| $\Delta$Pop $\rightarrow$ $\Delta$FF | Direct | *If* population increases, *then* forest fires increase |
| $\Delta$Pop $\rightarrow$ $\Delta$Trans-Ve | Direct | *If* population increases, *then* the number of vehicles increases |
| $\Delta$Pop $\rightarrow$ $\Delta CO_2$ | Direct | *If* population increases, *then* household appliances increase, which contributes to increase $CO_2$ emissions |
| $\Delta$Pop $\rightarrow$ $\Delta$Trans-Ve $\rightarrow$ $\Delta CO_2$ | Indirect | *If* population increases, *then* the number of transport vehicles increases, which will contribute to increase the $CO_2$ emissions |
| $\Delta$Pop $\rightarrow$ $\Delta$FF $\rightarrow$ $\Delta$LVC | Indirect | *If* population increases, *then* forest fires (intentional or no-intentional) increase, thus increasing the loss of vegetation cover. |
| $\Delta$Pop $\rightarrow$ $\Delta$Trans-Ro $\rightarrow$ $\Delta$LVC | Indirect | *If* population increases, *then* the roads of transportation increase, thus increasing the loss of vegetation cover |

The independent variable will, in most cases, be the population. This is because we are interested in knowing whether there is a relationship between the increase in population and the increase in the other involved variables (the dependent variables). We classify these relationships as direct and indirect. A direct relationship takes place when the population increase is directly linked to a key environmental variable. Meanwhile, an indirect relationship takes place when the population increase is linked to a key environmental variable through intermediate variables. Such is the case of the last three relationships shown in Table 4 below.

Based on the expressions depicted in column 1 of Table 4, the symbol "$\Delta$" represents in this work "an increase". Thus, when any variable is preceded by the symbol "$\Delta$", it means that this variable increases over time. The notation "A $\rightarrow$ B", means that A causes B. The meaning of the acronyms representing the variables used in Table 4 above have already exposed in Table 2 (Section 2).

The set of graphics corresponding to pairs of causal relationships in this work are calculated using the percentage increase of each variable involved in this study. The necessary data related to each variable to calculate the percentage increases are provided by Table 1 in Section 2.

The percentage increase is calculated by the following expression: $(V_{\text{current-year}} - V_{2000})/V_{2000}$. We show an example to illustrate how it works. We use in this example the data related to the variable of loss of vegetation cover (LVC). A special case is represented by the year 2000. If the $V_{\text{current-year}}$ is the one corresponding to the year 2000, then replacing the values in the preceding expression yields:

$(V_{\text{current-year}} - V_{2000})/V_{2000} = (90.4 - 90.4)/90.4 = 0$ (%); As expected, there was no increment, because the year 2000 is the reference.

For the $V_{\text{current-year}} = V_{2001}$; $((V_{2001} - V_{2000})/V_{2000}) = (201.5 - 90.4)/90.4 = 112.8\%$. That is, the variable of LVC increases 112.8% with respect to the LVC value in the year 2000.

For $V_{2002}$; $((V_{2002} - V_{2000})/V_{2000}) = (257.0 - 90.4)/90.4 = 184.29\%$. In this case, the variable of LVC increases 184% with respect to the LVC value in the year 2000.

The rest of the values of the percentage increase corresponding to a specific year are calculated in a similar way. Likewise, the values of percentage increases corresponding to the remaining variables involved in this study are calculated in a similar way. Table 5 below shows the percentage increase of each variable involved in this study.

**Table 5.** The percentage increase of each variable involved in this study.

| Year | ΔPop (%) | ΔCO$_2$ (%) | ΔTrans-Ro (%) | ΔFF (%) | ΔLVC (%) | ΔWat-Av (%) | ΔTrans-Ve (%) | ΔWas (%) | ΔAir-Quality (%) |
|------|------|------|------|------|------|------|------|------|------|
| 2000 | 0 | 0 | 0 | 0 | 0 | 0 | 0 | 0 | 0 |
| 2001 | 0.600 | 1.742 | 1.399 | 125 | 122.8 | 0 | 12.468 | 2.832 | 0 |
| 2002 | 1.204 | 5.634 | 1.399 | 475 | 184.29 | 0 | 20.501 | 5.229 | 8.191 |
| 2003 | 1.811 | 8.818 | 1.399 | 475 | 264.71 | 3.726 | 23.715 | 7.407 | 9.952 |
| 2004 | 2.422 | 4.749 | 2.848 | 475 | 348.34 | 4.081 | 28.535 | 14.597 | 6.153 |
| 2005 | 3.704 | 19.265 | 3.948 | 475 | 426.65 | 2.555 | 36.568 | 17.211 | 11.99 |
| 2006 | 5.778 | 25.370 | 3.948 | 475 | 509.84 | 27.999 | 60.668 | 19.390 | 16.58 |
| 2007 | 7.893 | 26.127 | 5.547 | 500 | 578.87 | 27.076 | 73.522 | 20.044 | 26.49 |
| 2008 | 10.051 | 29.709 | 23.788 | 529.16 | 654.20 | 27.289 | 86.375 | 20.915 | 16.86 |
| 2009 | 12.252 | 34.371 | 23.788 | 545.83 | 743.69 | 27.608 | 99.229 | 21.569 | 3.311 |
| 2010 | 14.269 | 37.036 | 49.325 | 554.16 | 832.85 | 29.489 | 118.509 | 29.847 | 4.662 |

The set of graphics showing the behavior of causal relationships involved in this study for the period 2000–2010 are shown in Figure 4 below. We use the method of simple linear regression to relate the independent variable that has an effect on the dependent variable. For each graphics, we show its corresponding trend, which is represented by the parameter $B_1$, as mentioned before. As already described, the interpolated straight line equation is $Y = B_0 + B_1X$; where X represents the independent variable, Y the dependent variable, $B_o$ is the value of Y when X = 0, and $B_1$ is the value of the slope of the interpolated straight line.

**Step 4.2.** *The method to build the causal network*. Finally, we use the casual relationships shown in Table 4 to build the causal network. In fact, the main product generated by Module 1 of the open system is the causal network. The method consists in linking the population node with the nodes representing the five key environmental variables. This method aims to determine how population increase influence on the increase of key environmental variables. We can see from Table 4 that the population node can be linked directly and indirectly with nodes associated with the key environmental variables. For example, the population node is linked directly with the nodes waste (Was), water-availability (Wat-Av), Air-Quality, and $CO_2$. However, population node can be also linked with the $CO_2$ node through the node representing the transport vehicles. That is, there are two ways to go from the population node to the $CO_2$ node. A similar case takes place to connect the population node with the LVC node. One way is connecting the population node with the LVC node through the forest fire node. The second way is to connect the population node with the LVC node through the transport routes (Trans-Ro) node. Therefore, two ways from the population node converge into the $CO_2$ and the LVC node. Based on a similar reasoning, there are five pathways associated with the key environmental variable that converge into the Global Environmental State node resulting from the population node. It means that we can measure the influence of the population node on the global environmental state through a set of environmental variables. Applying the rules exposed in the preceding lines to build the pathways that connect the population node with the key environmental variables, we built the causal network shown in Figure 5 below. We recall that the key environmental variables are $CO_2$, waste, water availability, loss of vegetation cover, and air quality.

**Step 5.** *Integrating Module 1 and Module 2.* The causal network is the input to Module 2, whose main function is to assess the current trends of the environmental pathways and the current trend of the environmental state index. In Module 2 one method is developed. It aims to assess the current trends of the environmental pathways and the current trend of the global environmental index. This assessment method uses addition and product operations between current trends related to the environmental pathways. To perform these assessments, we use the trend values of causal relationships shown in Table 6 below. Such values represent the slope value of the coefficient $B_1$ shown in the equations of the graphics depicted in Figure 4. The slope values can be represented by tangent, angular, and normalized values as shown in columns 2, 3, and 4 of Table 6.

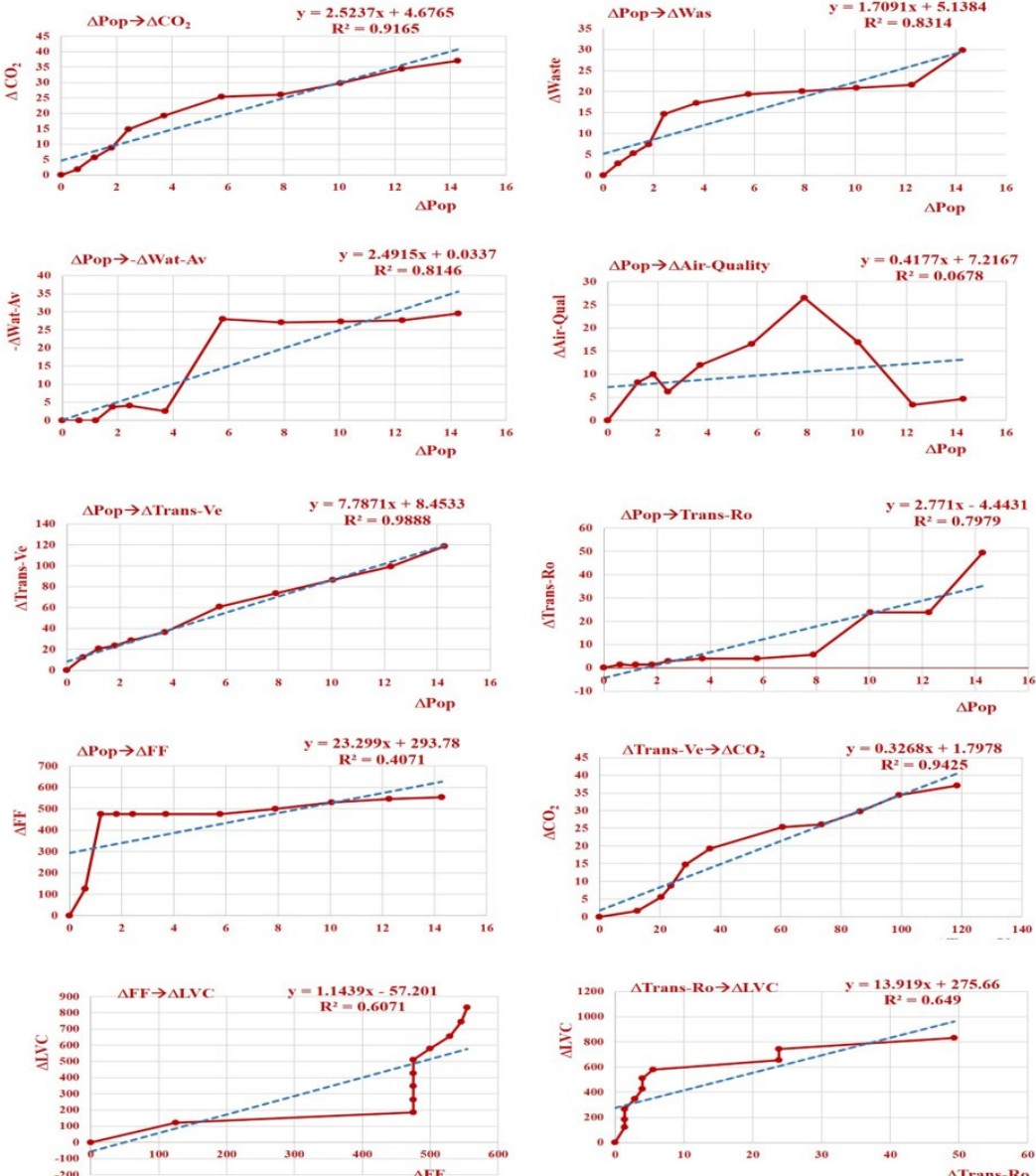

**Figure 4.** The relationships used in this study showing the observed points and the fitted straight lines along with their equation.

**Table 6.** The trend values of the causal relationships shown in Figure 4.

| Causal Relationships | Tangent Value | Angular Value | Normalized Value |
|---|---|---|---|
| $\Delta Pop \rightarrow \Delta CO_2$ | 2.5237 | 68.38° | 0.759 |
| $\Delta Pop \rightarrow \Delta Was$ | 1.7091 | 59.66° | 0.662 |
| $\Delta Pop \rightarrow \Delta Wat\text{-}Av$ | 2.491 | 68.13° | 0.757 |
| $\Delta Pop \rightarrow -\Delta Air\ Quality$ | 0.4177 | 22.67° | 0.251 |
| $\Delta Pop \rightarrow \Delta Trans\text{-}Ve$ | 7.7871 | 82.68° | 0.918 |
| $\Delta Pop \rightarrow \Delta Trans\text{-}Ro$ | 2.7718 | 70.16° | 0.779 |
| $\Delta Pop \rightarrow \Delta FF$ | 23.299 | 87.54° | 0.972 |
| $\Delta Trans\text{-}Ve \rightarrow \Delta CO_2$ | 0.3268 | 18.09° | 0.201 |
| $\Delta FF \rightarrow \Delta LVC$ | 1.143 | 48.83° | 0.542 |
| $\Delta Trans\text{-}Ro \rightarrow \Delta LVC$ | 13.919 | 85.89° | 0.954 |

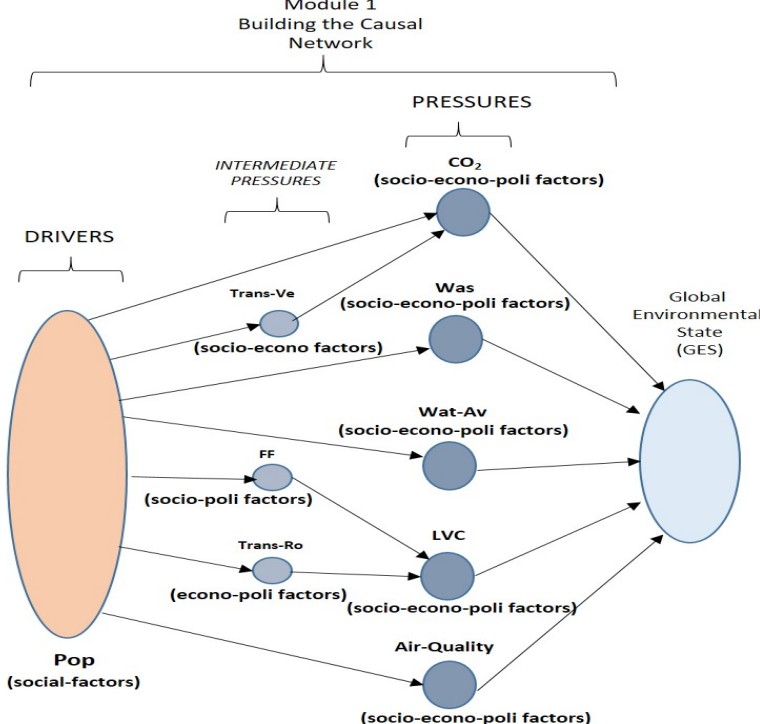

**Figure 5.** The causal network generated in Module 1.

The operations to obtain the current trends will use the normalized values situated in column 4 of Table 6.

**Step 5.1.** *Calculating the current trend of direct and indirect pathways*. The way of obtaining the current trend of direct pathways is simple. We take the value of the coefficient $B_1$ expressed in normalized values, such as those shown in Table 6. Whereas, the method for calculating the current trend of an indirect pathway is quite different. As an example, we show the case of the indirect pathways Path_LVC, which is composed of two paths that converge into the LVC node from the population node. Thus, the first path links the population node with the LVC node through the node FF. The second path links the population node with the LVC node through the node Trans-Ro. The expressions below show these two paths represented by cause-effect relationships:

- $(\Delta Pop \rightarrow \Delta FF) \wedge (\Delta FF \rightarrow \Delta LVC)$.
- $(\Delta Pop \rightarrow \Delta Trans\text{-}Ro) \wedge (\Delta Trans\text{-}Ro \rightarrow \Delta LVC)$

In both expressions above, the symbol "$\wedge$" plays a similar role than the operator "AND", which in turn represents a product operation when a transitivity situation takes place. The product operation is represented as usual by the symbol "**x**". Given that two pathways converge into the LVC node, the values of both pathways are added and then divided by 2 to obtain an average value. Thus, the final expression to calculate the current trend of Path_LVC is:

Path_LVC = $(((\Delta Pop \rightarrow \Delta FF) \times (\Delta FF \rightarrow \Delta LVC)) + ((\Delta Pop \rightarrow \Delta Trans\text{-}Ro) \times (\Delta Trans\text{-}Ro \rightarrow \Delta LVC)))/2$

Replacing each relationship by their normalized values shown in Table 6, the final current trend of the Path_LVC is:

Path_LVC = $((0.972 \times 0.542) + (0.779 \times 0.954))/2 = (0.526 + 0.743)/2 = 1.269/2 = 0.6345$.

A similar procedure is applied for the case of the pathways converging into the $CO_2$ node from the population node.

The results for each environmental pathway are shown in Table 7 below. Please note that the trend value of the Path_LVC was 0.6345 in the previous calculation. Meanwhile, the trend value of the same Path_LVC shown in Table 7 is 0.635. Hence, the difference between both values is 0.0005. Therefore, for practical reasons we have rounded 0.6345 to 0.635. We confirm that it is the case for other calculations.

**Table 7.** The pathways are derived from the causal network of Figure 5.

| The Pathways | Tangent Value | Angular Value | Normalized Value |
|---|---|---|---|
| *Direct Pathways* | | | |
| Path_Was = ($\Delta$Pop $\rightarrow$ $\Delta$Was) | 1.709 | 59.67° | 0.662 |
| Path_Wat-Av = ($\Delta$Pop $\rightarrow$ -$\Delta$Wat-Av) | 2.491 | 68.13° | 0.757 |
| Path_Air-Quality = (($\Delta$Pop $\rightarrow$ -$\nabla$Air-Quality) | 0.417 | 22.67° | 0.251 |
| *Indirect Pathways* | | | |
| Path_CO$_2$ = ((($\Delta$Pop $\rightarrow$ $\Delta$Trans-Ve)$\wedge$($\Delta$Trans-Ve $\rightarrow$ $\Delta$CO$_2$)) + ($\Delta$Pop $\rightarrow$ $\Delta$CO$_2$)))/2 | 0.915 | 42.48° | 0.472 |
| Path_LVC = ((($\Delta$Pop $\rightarrow$ $\Delta$FF) $\wedge$ ($\Delta$FF $\rightarrow$ $\Delta$LVC)) + (($\Delta$Pop $\rightarrow$ $\Delta$Trans-Ro) $\wedge$ ($\Delta$Trans-Ro $\rightarrow$ $\Delta$LVC)))/2 | 1.548 | 57.15° | 0.635 |

At this point, the trend values corresponding to links of relationships and the trend values of links connecting the nodes of the key environmental variables with the GES are displayed at the arcs of the causal network shown in Figure 5. Figure 6 shows the updated causal network with the trend values displayed at their corresponding arc.

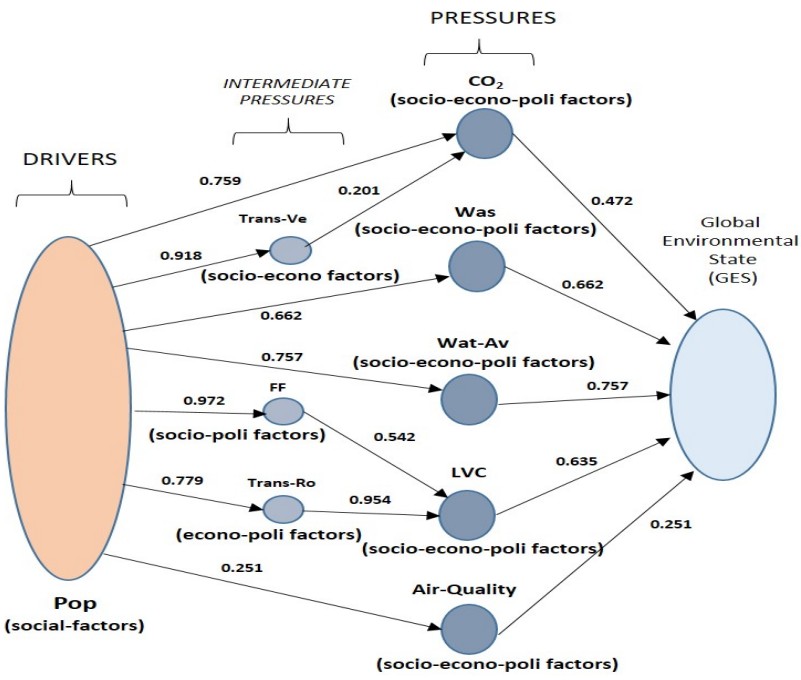

**Figure 6.** The causal network with the trend values of the causal relationship and the trend values of the pathways displayed at their corresponding arcs.

**Step 5.2.** *Building and calculating the current trend of the GES index.* In practical terms the GES (global environmental state) index is determined by the environmental pathways converging into the

GES node. In our case, there are five environmental pathways to be aggregated to obtain the GES index, as shown by the following expression:

$$GES = (Path\_Was + Path\_Wat\text{-}Av + Path\_Air\text{-}Quality + Path\_CO_2 + Path\_LVC)/5$$

Replacing the pathway values by their angular values shown in Table 7, the GES value yields:

$$GESa \text{ (angular value)} = (59.67° + 68.13° + 22.64° + 42.48° + 57.15°) = 250.07°/5 = 50.01°$$

The GES value can be converted to normalized and tangent values to give the three following values:

GESa (angular value) = 50.01°; GESn = 0.555 (normalized value); GESt = 1.192 (tangent value).

**Step 5.3.** *Assessing the impacts over time on the involved variables.* This simple method allows assessing the percentage increments of the involved variables in this study which depend on the population increase during the period 2000–2010. Column 4 of Table 8 below shows such percentage changes. The calculation is made using the following expression:

$$\text{Percentage diff}_{(2010\text{-}2000)} = ((V_{2010} - V_{2000})/V_{2000}) \times 100$$

**Table 8.** Impacts on the involved variables in this study. Percentage values of the year 2010 with respect to the year 2000.

| The Involved Variables | Year 2000 | Year 2010 | Percentage Difference between the Year 2010 and the Year 2000 |
|---|---|---|---|
| Population (persons) | 1,555,296 | 1,777,227 | 14.269% |
| $CO_2$ emissions (Gg) | 2816.2 | 3859.22 | 37.036% |
| Waste (tons) | 459,000 | 596,000 | 29.847% |
| Loss of water availability ($m^3$/person) | 2818 | 1987 | 29.489% |
| Loss of vegetation cover (ha) | 90.4 | 843.3 | 832.85% |
| Air quality (mass/$m^3$) | $1.01603 \times 10^{-8}$ | $1.0634 \times 10^{-8}$ | 4.662% |
| Vehicles of transport (number of vehicles) | 155,600 | 340,000 | 118.509% |
| Transport routes (km) | 2001 | 2986 | 49.225% |
| Forest fires (adults trees in hectares) | 12 | 78.5 | 554.16% |

We illustrate this calculation with the example of the LVC (loss of vegetation cover)

$$LVC_{2010} = 843.3 \text{ (ha)}; LVC_{2000} = 90 \text{ (ha)}$$

$$\text{Percentage diff}_{(2010\text{-}2000)} = ((843.3 - 90.4)/90.4) \times 100 = 8.3285 \times 100 = 832.85\%$$

The percentage difference of the involved variables in this study can serve for two purposes: (1) to evaluate what variables have been the most or the least damaged among the set of the involved variables; (2) to verify whether the relationship between the population increase and the pressure variables is linear or non-linear.

Up to this point, Module 1 and Module 2 are already connected as illustrated in Figure 7 below.

**Step 6.** *Selecting environmental management actions.* The assessment of the global environmental state index and the environmental pathways along with the assessment of impacts on the key environmental variables are the input data to Module 3. These assessments serve as support to decision-makers to select environmental management actions aimed at improving the environmental state. Until now, we have connected Module 1 and Module 2. In this step Module 1 and Module 2 will be connected with Module 3 to complete the Open System. This step requires the data depicted in Table 3 of the section of Materials. In this table a set of potential environmental management actions are proposed as candidates to be implemented to improve the current trend of the environmental state.

The method to select the environmental management actions takes into account the assessments of the current trends of both the environmental pathways and the current trend of the GES. In addition, a method aimed at situating the current trends of the environmental pathways and current trend of the GES within regions at different level of risk is required. Therefore, such method requires the definition of different regions where trend values can fall inside. Justification of this method: both the trend values of environmental pathways and the GES index can become risky trends as their trend values are getting closer to 90°. This fact can be also interpreted as follows: the independent variable exerts strong effects on the dependent variable in the neighborhood of 90°. On the contrary, if the trend value of pathways and/or GES index tends to 0°, then minimal risks take place or the effects of the independent variables on the dependent one would be minimal. We have to recall that the maximum angular value of a trend is 90° and the minimum value is 0°. Based on these arguments, we would need to define a region at very high risk and a region at very low risk. However, the definition of just two regions for such a wide space (0° to 90°) may result in their not being practical enough. Hence, we have also defined regions at intermediate risks. The classification of current trend values at different levels of risk will provide decision-makers with useful information in the processes to select adequate environmental management actions. Based on these arguments, we propose five regions at different level of risk to situate the trend values, as shown in Table 9 below.

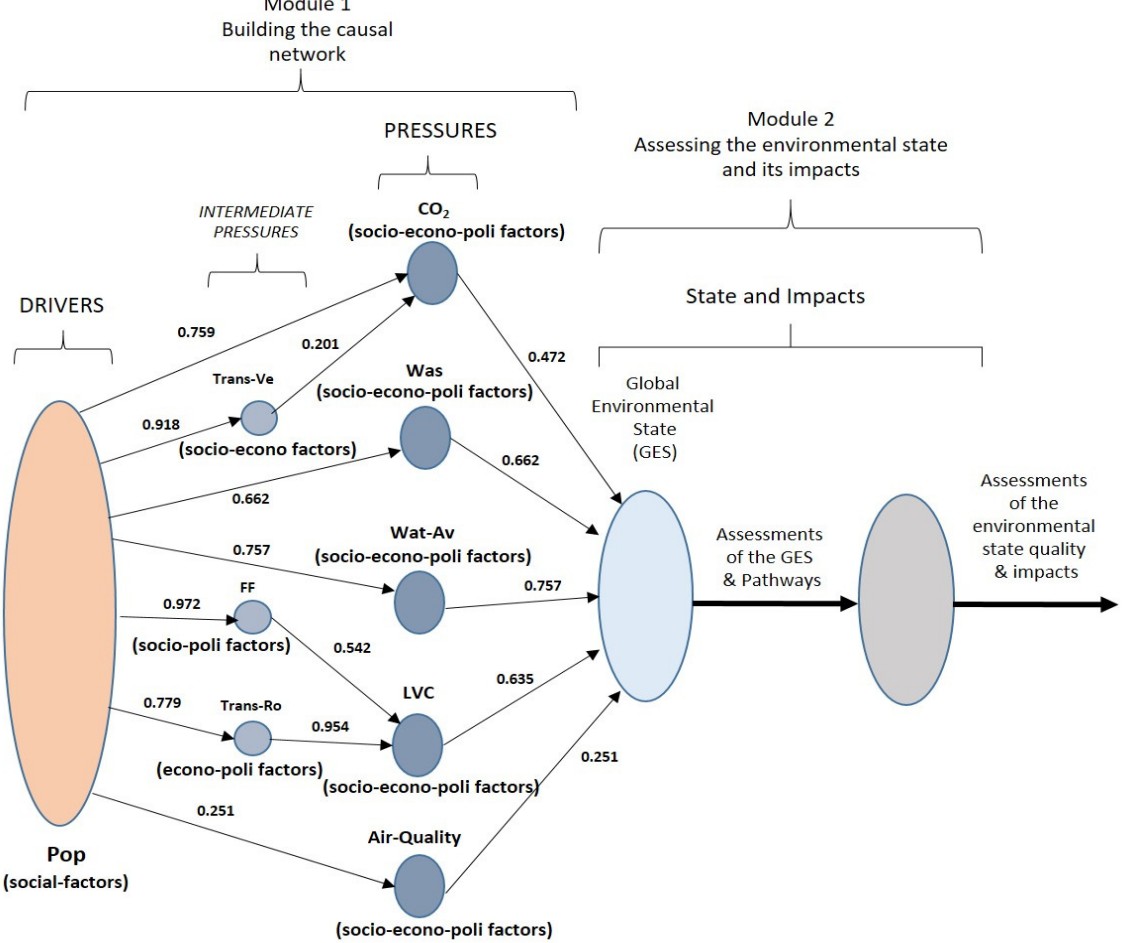

**Figure 7.** Module 1 is connected with Module 2. The causal network has been built and the assessments of environmental pathways and the GES have been performed.

As we can see in Table 9, we have defined five regions at risk. However, they could be less than five or more than five. If we define 10 regions, for instance, their ranges could tend to be very narrow and consequently the task for searching adequate environmental management actions applicable to 10

different ranges could result harder an non practical. Another point to be highlighted refers the last range of Table 9, if it is expressed in angular values, this range is [80°, 90°], and in tangent values is [5.6718, ∞]. This range [80°, 90°] has an exponential behavior when it is expressed in tangent values as shown by the following sample of calculations: tg(82°) = 7.11536; tg(84°) = 9.5143; tg(86°) = 14.30006; tg(87°) = 19.08113; tg(88°) = 28.63625; tg(89°) = 57.28996; tg(89.5°) = 114.58886; tg(89.7°) = 190.9841; tg(89.99°) = 5729.5778; tg(89.99999°) = 5,729,577.951; . . . . . . . . . . . . ; tg(90°) = ∞. We can see that as the angular values are getting closer to 90°, any small variation between two neighbor angular values close to 90° tends to represent a huge difference, when it is expressed in tangent values. Based on these calculations, we can approximate the first four ranges to a linear behavior and the last range to an exponential behavior. This is the reason the last range [80°, 90°] is different than the others.

**Table 9.** Risk regions where values of the pathways and GES trends could fall.

| Regions at Risk | Ranges of the Trends in Angular Values | Ranges of the Trends in Tangent Values |
|---|---|---|
| Very low risk | [0°, 20°) | [0, 36,397) |
| Low risk | [20°, 40°) | [0.36397, 0.83909) |
| Medium risk | [40°, 60°) | [0.83909, 1.73205) |
| High risk | [60°, 80°) | [1.73205, 5.6718) |
| Very high risk | [80°, 90°] | [5.6718, ∞] |

We show an example to illustrate how the method to select environmental management actions works. For this example, we take the values of the environmental pathways and the GES already calculated. We show their current trend values in terms of angular values along with the region at risk to which they fall inside: GES = 50.01° (region at mid-risk); Path_CO$_2$ = 42.48° (region at medium-risk); Path_Was = 59.67° (region at medium risk); Path_Wat-Av = 68.13° (region at high risk); Path_LVC = 57.15° (region at medium risk); Path_Air-Qual = 22.64° (region at low risk).

Two aspects should be considered to select the environmental management actions: (1) the regions at risk assigned to the environmental pathways based on the value of their current trend; (2) where the current trend value is located within the assigned range. For instance, Path_Wat-Av is located in the region at high risk, but it is practically in the middle of the range. Whereas, both Path_LVC and Path_Was are located in the region at mid-risk, but very close to the beginning of the region at high risk. These two aspects should be taken into consideration by decision makers to select the environmental management actions from those described in Table 3 of the Materials section.

We would like to point out that both the environmental key variables and the environmental management actions have been chosen based on the guidelines of an Outlook to 2030 made by the OECD [7]. This outlook describes a set of environmental scenarios for the year 2030 that could result depending on the performance of the implemented management actions.

These scenarios have been labeled with a green light (good scenarios), a yellow light (more or less good scenarios), and a red light (bad scenarios). The types of implemented management actions and the way they are implemented will determine what scenarios would be reached. The environmental issues chosen by this outlook are associated with five key environmental variables CO$_2$ emissions, Biodiversity, Water, Waste, and Air Quality. The key environmental variables of this study match with the environmental issues proposed by the OECD-Outlook to 2030.

Up to this point, we have already connected the three modules to complete the Open System. Figure 8 shows the Open System connecting Module 1, 2, and 3.

**Step 7.** *Building the Closed System*. The performance of the implemented management actions is assessed in the feedback loop to determine whether the implemented management actions are reinforced, modified, and/or eliminated. The feedback loop can impact on any factor of the system, and consequently on any of the modules, and on their models or methods, of the open system. Figure 9 shows the Closed System.

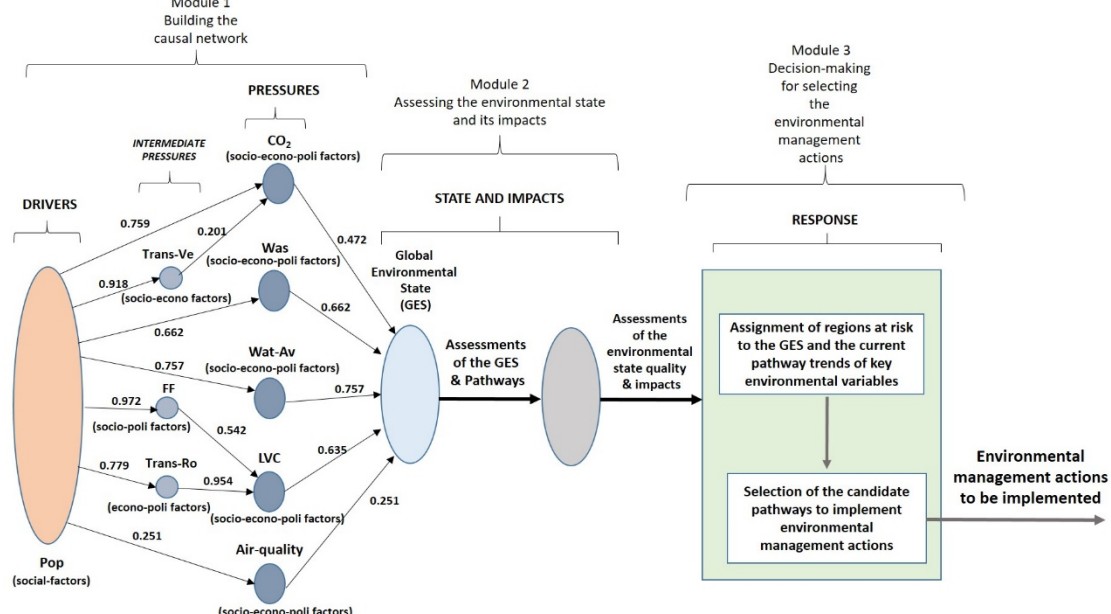

**Figure 8.** The Open System.

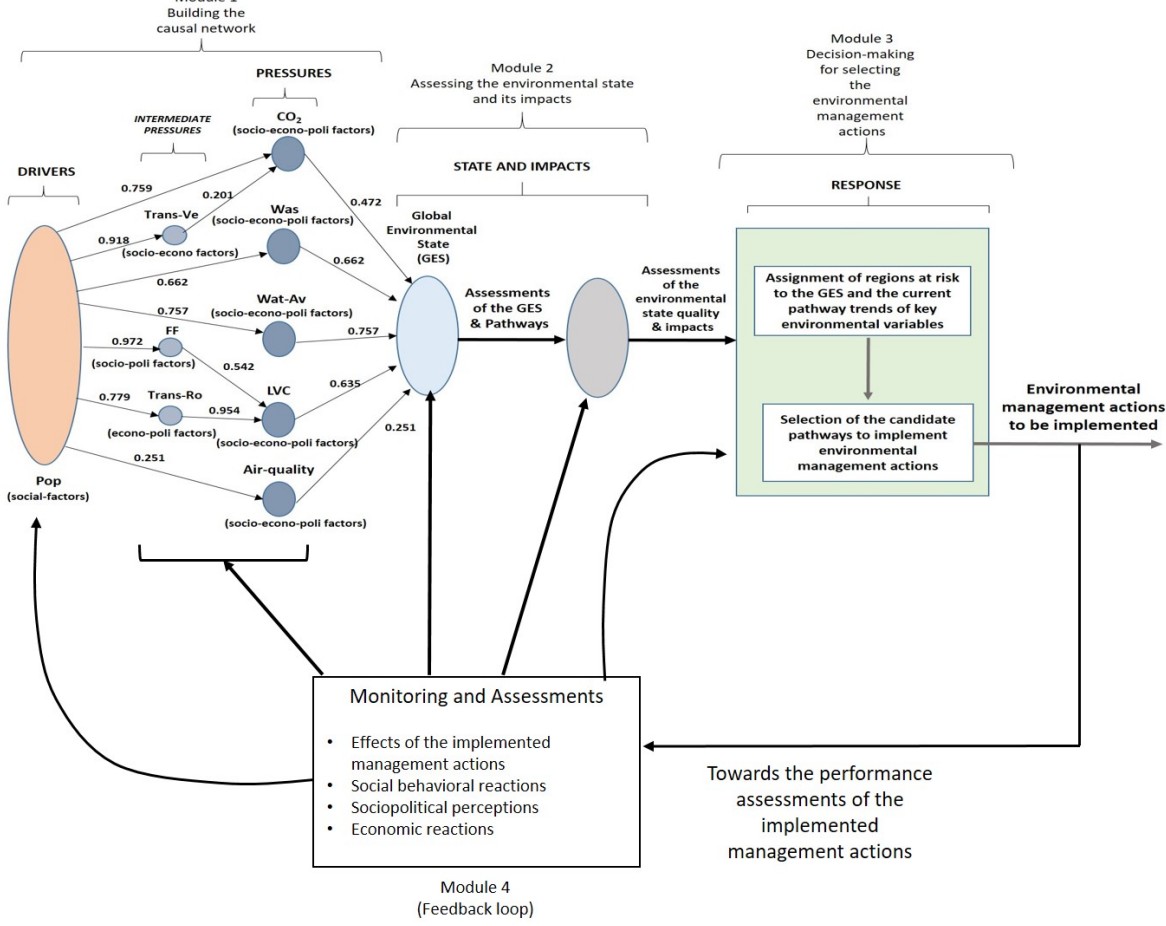

**Figure 9.** The Closed System.

## 3. Discussion of Results

*On the proposed methodology*. It is important to mention that the methodology presented in this work has not been previously published, but it has been proposed in this paper. This methodology aims

to build a conceptual framework supported by a systems thinking approach by including a feedback loop to achieve a closed system. We aimed at building a closed system to pave the way towards the development of a sustainable environmental management system. Like most of the methodologies of this nature, it should be applied to real cases, as an important part of it. Thus, this methodology was applied to the case of the state of Morelos, Mexico using real data of the period 2000–2010.

***On the analysis of multiple factors related to human activities and nature that affect the environmental state of the region treated in this paper.*** Such multifactorial interactions bring about complex dynamic processes that are hard to understand and consequently hard to assess, thus requiring the support of the theory of complexity and by considering that natural and social systems are dynamic and non-linear [2].

In this paper, we have dealt with population increase (a social factor) as a driving force interacting with nature through pressure variables causing effects on the environmental state quality.

We have defined, in this work, a set of key environmental variables related to water availability, solid waste generation, $CO_2$ emissions, loss of vegetation cover, and the air quality represented by the production of $PM_{2.5}$ particles. These variables match the key environmental variables suggested by the OECD-Outlook to 2030 [7]. Other variables such as forest fires, transport routes and transport vehicles that depend importantly on the population increase were included in the study. It is important to mention that these last three variables exert important effects on the five key environmental variables mentioned before.

We point out that we dealt, in this work, with variables whose information was available in the state of Morelos, Mexico, during the period 2000–2010. A set of factors belonging to social, economic, and environmental domains that are involved in human-nature interactions affects the variables considered in this work. For instance, as the population and economic development increase (socioeconomic factors), the generation of solid waste increases; water availability decreases because of water catchment decline (a nature factor) due to intentional change of land use (a social factor) and intentional (a social factor) or non-intentional (a nature factor) forest fires; important effects of the increase of $CO_2$ emissions are due to human consumption (a socioeconomic factor), industrial development (an economic factor), increase of the vehicular fleet (socioeconomic factors), and the decline of $CO_2$ sequestration, which can be related to the loss of forests (a nature factor and a social factor), among the most important factors; the increase of the loss of vegetation cover is being caused constantly by intentional and/or non-intentional forest fires, construction of new industrial and urban areas, construction of new transport routes and the change of land use (social, economic, and nature factors), among the most important; the air quality decreases due to gases emitted by the increment of household appliances, industries, and transport vehicles (socioeconomic factors).

***On the support of the systems thinking approach in the construction of a closed system and a conceptual framework.*** The fact of building a closed system supported by a systems thinking approach allowed defining and connecting two important parts of the process: the open system and the feedback loop. The closed system was built at high level of abstraction. At this level, we describe the available information at the input of the open system, the final product at its output, which is represented by a set of environmental management actions. The input of the feedback loop, therefore, is represented by the monitored data from the implemented management actions. While its output is represented by the recommendations to improve the performance of the system (see Figure 2).

One of the advantages of building a closed system supported by a systems thinking approach was to build a conceptual framework and paving the way towards the development of a sustainable environmental management system. On the one hand, because the structure of the conceptual framework allows distributing and interconnecting the different nature of factors in their corresponding category. Such categories are related to drivers, pressure variables, assessments of environmental state and impacts, decision-making process, and feedback loops. On the other hand, the monitoring and assessments of the implemented management actions through the feedback loop contribute favorably towards the development of a sustainable environmental management system.

***On the Open System.*** It was analyzed in a similar way to a process, thus with the following important advantages: (1) we can specify the final product or output in function of the available resources at the input of the systems process; (2) we may have available an adequate structure to define the different input-output sequences and their associated methods to achieve the objectives of each phase.

Based on this method, the open system has been defined by three modules connected as a sequence of inputs and outputs. The definition of inputs and outputs of each phase of the process allowed defining the required methods to achieve the specified outputs in function of the inputs.

***On the Module 1 of the Open System and its associated methods.*** In the first module, we develop the method to build the causal network using causal relationships. In the development of the causal network, we looked for building sequences of causal relationships linking the population node with the key environmental variables. Such sequences of casual relationships were named key environmental pathways, each one associated with a key environmental variable ($CO_2$, Waste, Water, Loss of Vegetation Cover, and Air-Quality). The conceptual framework using a systems thinking approach facilitated the construction of the causal network, from which a set of environmental pathways was built. The five environmental pathways served, in turn, to determine the global environmental state (GES) index.

In the present study, the current trend value of GES was 0.555, expressed in normalized values or 50.01°, expressed in angular values. Therefore, the GES value falls within the region at mid risk (see Table 7). Based on this value, we were interested in knowing the contribution of each environmental pathway to this global value. The current trends of pathways with the highest value corresponded to Path_Wat-Av (68.13°), Path_Was (59.67°), and Path_LVC (57.15°). As we can see in Table 6, both the percentage value of solid waste and the loss of water availability increased two times more than the percentage value of the increase in population during the period of time considered. We also calculated that the percentage increase of the LVC was approximately 28 times more than the percentage increase of water availability and waste. Based on this calculation, we would expect that the region at risk that corresponds to the trend value of the LVC pathway would be a "region at very high risk". Nevertheless, it was not the case. We know that both the Path_Was and Path_Wat-Av are direct pathways. That is, the population node connects directly with both the waste node and water availability node. Meanwhile, the LVC pathway is an indirect pathway, which means that the population node connects the LVC node through intermediate nodes, in this case, the transport node and the forest fire node. Arithmetic operations are necessary to obtain a final result related to the Path_LVC. Hence, we calculated the final current trend value of the Path_LVC. The expression to calculate the final trend value of the Path_LVC is shown below:

Path_LVC = ((($\Delta$Pop → $\Delta$FF) ∧ ($\Delta$FF → $\Delta$LVC)) + (($\Delta$Pop → $\Delta$Trans-Rou)) ∧ ($\Delta$Trans-Rou → $\Delta$LVC)))/2

Replacing the values of the causal relationships involved in the Path_LVC (see Table 6), it yields:

Path_LVC = (0.972 × 0.542 + 0.779 × 0.954)/2 = (0.526 + 0.743)/2 = 0.634 (a normalized value)

If the Path_LVC is expressed in angular values, it yields:

Path_LVC(degrees) = 0.634 × 90 = 57.06°.

What is important to point out is that the final value of Path_LVC depends on the number of intermediate nodes between the population node and the LVC node. Meaning also that as the number of intermediate nodes between the head node and the tail node increases, the final value of the sequence will tend to represent a very low value. This is because, the value of two relationships linked by intermediate relationships is performed by the product operation. As we know, the product of two operands, whose values are less than 1, will always be smaller than the smallest value of the two operands. Based on the preceding expression, the relationship value of ($\Delta$Pop → $\Delta$FF) was very high,

with a normalized value of 0.972 or 87.48° expressed in angular value (this value is situated in the region at very high risk), see Table 9. Meanwhile, the relationship ($\Delta$FF → $\Delta$LVC) was 0.542 (normalized value) or 48.78° expressed as angular value (this value is situated in the region at medium risk). The final result of the dependence relationship between the population increase and the LVC increase through the FF increase is 0.526 (normalized value) or 47.34° expressed as angular value (this value is situated in the region at medium risk). A similar situation takes place with the causal relationship ($\Delta$Pop → $\Delta$LVC) through the construction of transportation routes (Trans-Rou), whose final value was the product 0.779 × 0.954 = 0.743. Therefore, the average value of the Path_LVC is (0.734 + 0.526)/2 = 0.6342. Due to the transitivity relationships the current trend value of the relationship between LVC and the population increase has been 0.634 (a rounded normalized value) or 57.06° (angular value). As a result, this value is situated in the region at medium risk with a trend towards the region at high risk.

A similar situation occurs with the causal relationship ($\Delta$Pop → $\Delta$CO$_2$) through the increase of the number of transport vehicles ($\Delta$Trans-Ve). The expression that represents this relationship is shown below:

$$\text{Path\_CO}_2 = (((\Delta\text{Pop} \to \Delta\text{Trans-Ve}) \wedge (\Delta\text{Trans-Ve} \to \Delta\text{CO}_2)) + (\Delta\text{Pop} \to \Delta\text{CO}_2)))/2$$

$$\text{Path\_CO}_2 = (0.918 \times 0.201 + 0.759)/2 = (0.1845 + 0.759)/2 = 0.4717$$

As we can see from the expression above, the population node converges into the CO$_2$ node by two paths. One direct path from population node to the CO$_2$ node, and one indirect path through the Trans-Ve node. In this case, the product of the relationship ($\Delta$Pop → $\Delta$Trans-Ve) = 0.918 (a normalized value situated in the region at very high risk), with the relationship ($\Delta$Trans-Ve → $\Delta$CO$_2$ = 0.201 (a normalized value situated in the region at very low risk) yields (0.918 × 0.201) = 0.1845, or 16.60° expressed in angular values. It means that this indirect path is situated in the region at very low risk. Meanwhile, the normalized value of the direct path from population to CO$_2$ was 0.759 (normalized value) or 68.31°, expressed in angular values. It means that it is situated in the region at high risk. The average normalized value between these two paths was represented by (0.1845 + 0.759)/2 = 0.471 expressed in normalized values or 42.45° expressed in angular values. It means that the Path_CO$_2$ is situated in the region at medium risk. We can see that the product operation expresses coherently the fact that the influence of the population increase on the increase of a key environmental variable is reduced. It happens when they are linked through intermediate relationships, which is supported by the transitivity rule.

The low value of the relationship ($\Delta$Trans-Ve → $\Delta$CO$_2$) = 0.201 or 18.09°, whose angular value is situated within the region at very low risk, can be explained by the implementation of the public policy related to the vehicle verification program. This program aims to control the emission of greenhouse gases in both Mexico City and the state of Morelos. We need to point out that there is an important movement of vehicles between Mexico City and Morelos, mainly for business reasons; they are neighboring cities. This traffic increases exponentially at the weekends and during vacation periods because Morelos is a tourist attraction (Cuernavaca is the capital of the state of Morelos and is called "the city of eternal spring"). Also, an important number of residents from Mexico City have weekend houses in Morelos. Moreover, Morelos is situated between Mexico City and Acapulco (one of the most famous beach destinations in Mexico). Thus, to travel by car from Mexico City to Acapulco it is necessary to pass through Morelos. These aspects could contribute, apparently, to increasing CO$_2$ emissions. However, thanks to the vehicle verification program designed to control the emission of greenhouse gases, the increase in transport vehicles belonging to the state of Morelos and Mexico City contribute less than other factors to the increase of CO$_2$ emissions.

The trend value of the relationship between population and air quality was 0.4177 (a tangent value) or 22.67° (an angular value), whose value belongs to the region at low risk. This value seems to be non-significant. However, it is important to mention that this population increase corresponds to the rural population, which is transported mostly in public transport.

***On the Module 2 and its Methods.*** In Module 2, we have built and also calculated the current trend of the GES index as an average of the environmental pathways. This global index or aggregated indicator provide us with a global assessment of the environmental state through a value that can vary from 0 to 1 (normalized values) or from 0° to 90° (angular values). This value can result challenging to interpret because it represents several indicators at the same time. However, it could be simpler to be interpreted when it is represented in terms of the region at risk to which it belongs. For instance, if the current trend value of the GES was 0.555, expressed in normalized values or 50.01°, expressed in angular values, then it can be hard to have a clear idea whether this value is a good signal or not of the environmental state. Nevertheless, if the current trend value of the GES is expressed in terms of the regions at risk to which it is situated, which is the region at mid risk, the idea about the global environmental state (GES) becomes easier to interpret.

***On the percentage difference between the average value of the population and the involved variables for the period 2000–2010.*** As we can see in Table 8, almost all of the percentage values of the involved variables in this study, for the period 2000 to 2010, increased more than the percentage value of the population increase. The only variable whose percentage value increased less than the population was Air-Quality. Other percentage values of variables such as LVC increased more than 28 times the percentage value of solid waste and the loss of water availability. Meanwhile, the percentage values of solid waste increase and the water decrease were more than two times the percentage value of the population increase. This fact gives us an idea about what relationships between the population increase and the increase of key environmental variables could result linear and non-linear.

However, what is important in this study is not the increase of each variable individually, but the trends (upwards or downwards) of relationships between variables. Moreover, one of the main purposes of this study is to assess the current trends of environmental pathways associated with the key environmental variables. We recall that the environmental pathways are composed of sequences of causal relationships linking the driving force variable (the population increase) with the environmental key variables. In other words, we aim to know the behavior of the key environmental variables vis à vis the population changes over time. In addition, the analysis based on the key environmental pathway enable us to identify what key environmental variables contribute the most or the least to the current trend of the GES.

***On the Module 3 and the selection of environmental management actions.*** At the end of the Open System, Module 3 will select the management actions based on the assessments made in Module 2. As an example, we select the environmental management actions and the angular values of the current trends concerning the pathways described in Table 7: Path_Was = 59.67° (region at medium risk); Path_Wat-Av = 68.13° (region at high risk); Path_Air-Qual = 22.64° (region at low risk); Path_CO$_2$ = 42.48° (region at medium-risk); Path_LVC = 57.15° (region at medium risk). Thus, the current trend of the GES is 50.01° (region at medium risk). As already mentioned, three pathways resulted the candidates to implement environmental management actions to improve the environmental state: Path_Wat-Av, Path_Was, and Path_LVC. We can see that Path_Wat-Av is situated in the region at high risk, but its current trend value tends toward the region at very high risk; and both the Path_Was and Path_LVC are situated in the region at mid risk, but they tend towards the region at high risk. Thus, these three paths are serious candidates to apply environmental management actions to reduce the current GES trend. Based on these assessments, decision-makers will select the environmental management actions from the list depicted in Table 2 (section of Materials) that they consider as potentially implementable. Therefore, the output of Module 3 will represent a list of implementable management actions.

As mentioned in the subsection that dealt with the selection of environmental management actions, two aspects should be taken into consideration by decision makers to select the environmental management actions: (1) the regions at risk assigned to the environmental pathways based on the value of their current trend; (2) and where the current trend value is located within the assigned range. However, other aspects, such as those related to sociopolitical, socio-economical, and technical

feasibility should be taken also into account. These aspects bring about priorities that depend on each geographical region where the environmental management actions will be implemented. As expected, the priorities of region X are not necessarily the same for region Y. We can conclude that the proposed methodology aims to support decision makers with assessments of the environmental state and the current trends of environmental pathways to guide the selection of environmental management actions. However, other important aspects could play a relevant role in the final selection.

*On the Feedback loop.* The implementation of management actions should be monitored and assessed to measure their performance through a feedback loop performed in Module 4, thus building a Closed System. Based on the performance of the implemented management actions decision makers can reinforce, change or eliminate the implemented management actions. The feedback could produce effects on any of the Modules 1, 2, and 3 and their methods.

In this work, we have assessed the effects of two implemented environmental management actions aimed at reducing deforestation in Morelos. The first one is called "natural protected areas" (NPA) and the second one "payment for environmental services" (PES). The implementation of these actions would have had positive effects by reducing the trend of the loss of vegetation cover. However, both measures failed. The performance assessment of these actions is a way of monitoring its effects through a feedback mechanism whose implementation converts the open system in a closed system as shown in Figure 9.

We analyze the results of the implemented management actions related to NPA and the PES to protect forestry areas. Both are public policy measures to mitigate and control the loss of vegetation cover that were significantly caused by intentional deforestation to change the land use. We first deal with the case of the NPA. Table 10 below shows the real data and the value of the percentage increase for both the variable LVC and NPA.

**Table 10.** Data related to NPA and the LVC during the period 2000-2010. Columns 3 and 5 show the percentage values of both variables.

| Year | LVC (ha) | Percentage Increase LVC | NPA (ha) | Percentage Increase NPA |
|------|----------|------------------------|----------|------------------------|
| 2000 | 90.4 | 0 | 120,020.31 | 0 |
| 2001 | 201.5 | 112.8 | 120,020.31 | 0 |
| 2002 | 257.0 | 184.29 | 120,020.31 | 0 |
| 2003 | 329.7 | 264.71 | 120,020.31 | 0 |
| 2004 | 405.3 | 348.34 | 120,020.31 | 0 |
| 2005 | 476.1 | 426.65 | 120,020.31 | 0 |
| 2006 | 551.3 | 509.84 | 120,020.31 | 0 |
| 2007 | 613.7 | 578.87 | 120,020.31 | 0 |
| 2008 | 681.8 | 654.20 | 128,397.33 | 6.979 |
| 2009 | 762.7 | 743.69 | 128,397.33 | 6.979 |
| 2010 | 843.3 | 832.85 | 128,656.26 | 7.195 |

Based on Table 10, we verify that the measure NPA was not applied during the period 2000–2007. This measure was applied until the year 2008 protecting only 8377.02 (ha). This measure was not applied in the year 2009. In the year 2010 the measure was applied to protect only 258.93 (ha) with respect to the year 2009. Thus, a total of 8635.95 (ha) was protected representing only 7.195% with respect to the protected area until the year 2000. In conclusion, the effects of the NPA measure on the loss of vegetation cover was not significant during this period due to the nonexistent application in the first 7 years and to a poor application during the period 2008–2010. We can verify that a feedback analysis provides decision makers with meaningful information derived from monitoring and assessments of the performance of the implemented management actions to reinforce or change them. This is one of the powerful advantages of building closed systems instead of open systems.

As usual, the lack of data for the case of payment for environmental services (PES) prevented a minimal analysis. We confirm that another way of avoiding an assessment of the performance of an

implemented environmental management action is the lack of quality in the available data and/or missing information, such is the case for the PES program applied to the state of Morelos, during the period 2000–2010. However, we make some important observations about this measure to clarify the reasons why its application was unsuccessful. The PES program provides landowners with annual payments to maintain the forest cover based on a single contract for 5 years. The payments represent approximately 20 USD/ha [68], instead of 400 USD/ha, which is the estimated cost required to avoid changing land use [69]. The preceding reason highlights acts of corruption, because the payments were made but the actions were not executed because of a difference of 380 USD/ha. These facts show that the agreements between landowners and the official institutions charged with the application of these measures are not formal and consequently impossible to monitor and assess, because of the lack of information.

We can conclude in simple words that a formal framework is required to implement management actions, which should be monitored and assessed within a context of a closed system by taking into account the social, political and environmental factors. Otherwise, the implemented management actions will remain within an open system context without the possibility of assessing their performance. We can summarize that the causes of non-successful implemented actions are the following: the environmental management action related to NPA was not applied regularly, moreover, it was not applied during several years (8 years/11 years); there were no formal agreements specifying compromises and obligations from both parts; lack of information, thus avoiding the assessment of the performance of the implemented environmental management actions. As we can see, sociopolitical factors can hinder the good performance of the implementation of management actions.

Some aspects of nature resilience could be reinforced if adequate management actions are implemented. Otherwise, the resilience of nature will continue to be threatened by human activities. Thus, sustainable human-nature interactions should be considered as important strategies to be generated by decision-makers. Such strategies need to be supported by an adequate understanding and assessment of the interactions that take place between humans and nature.

## 4. Conclusions

In this work, we have proposed that the systems thinking approach can be a meaningful support to build adequate conceptual frameworks that facilitate the understanding of the dynamic processes brought about by multifactorial human-nature interactions that affect the state of the environmental quality. In addition, this approach can be a useful aid to design closed systems thus providing developers with a structural vision of the whole system under study, instead of partial views.

The conceptual framework built within a context of a systems thinking approach represented a useful support to facilitate the achievement of three important issues; (1) the understanding of the dynamic processes involved in the environmental system under study, thus resulting in a better support for decision makers; (2) to place or situate drivers, stressors and pressures variables within their correct category, which in turn, facilitates the construction of causal networks and the construction of environmental pathways derived from them; (3) the analysis and assessments of environmental pathways as well as the analysis and assessments of the performance of the implemented management actions within a context of a closed system.

Based on the advantages described above, we are paving the way towards the development of sustainable environmental management systems.

The model was applied to the case of human activities related to the population increase that interact with nature through pressure variables causing damages to the state of the environment in the state of Morelos, Mexico. The study was carried out with real data compiled during the period 2000–2010. The driving force variable was the population increase causing effects on key environmental variables that match with those described by an OECD-Outlook to 2030 [7].

The conceptual framework facilitated the construction of a causal network, from which a set of environmental pathways was built; each one is associated with one of the five key environmental

variables. The five environmental pathways were used to determine a global environmental state index. The assessment of the environmental pathways and the global state of the environment provide useful information for decision makers to select potential environmental management actions aimed at reducing risky trends of the environmental pathways.

The current results suggest further research is necessary to reinforce the understanding of the complexity brought about by multiple interactions between drivers and pressure factors that could damage human-nature interactions, thus exerting negative effects on the environmental state.

**Author Contributions:** F.R.-Q. proposed the main conceptual ideas related to this work. H.S.-N. (Hugo Saldarriaga-Noreña) participated in the generation and analysis of the data yielded by the HYSPLIT4 model. F.R.-Q., H.S.-N. (Hugo Saldarriaga-Noreña), and H.S.-N. (Héctor Sotelo-Nava) designed the structure of the paper. F.R.-Q., H.S.-N. (Hugo Saldarriaga-Noreña), H.S.-N. (Héctor Sotelo-Nava), and E.T.-S. participated in the analysis of environmental pathways.

**Funding:** This research was funded by the program CONACyT-FOMIX of the Mexican Government of the State of Morelos, under the project No. 189949.

**Acknowledgments:** We thank anonymous reviewers and Rosalind Pearson Hedge for their comments and suggestions that improved our manuscript.

**Conflicts of Interest:** The authors declare no conflict of interest.

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
