# Peer review of "Assessing the Environmental Quality Resulting from Damages to Human-Nature Interactions Caused by Population Increase: A Systems Thinking Approach"

_sustainability, doi:10.3390/su11071957_

Round 1

Reviewer 1 Report

Abstract: the problem statement and importance of this issue needs to be clarified in abstract. The structure of abstract need to be revised.

Line 30-31: although it refers to result but the actual result of the study needs to be covered as it most like the suggestion for further research.

Introduction: The definition of some terms needs to be clarified. Although they might seems clear but you should clarify what they mean in your study such as “human-nature interactions”. The previous literature about the issue mentioned in line 36-43 needs to be added.

Section 1.1: it would be better to add some statistical numbers and tabulate the information or create a graphic to communicate the information more clear and effective.

The structure of introduction needs to be revised in terms of the hierarchy of information and add missing paragraphs. For example, 1.2 start discussing the problem of modelling without giving the literature about different methods and modelling that have been used since now and how it works. Another example 1.3. Also, the titles of some sections are not in line with their content “1.2”.

The aim of the study and method that have been used are mentioned inconsistent in the introduction (needs to be revised).

2.1 Materials: the paragraph needs to be rewritten and add more detail about the factors and data. The variables and factors need to be introduced clearly and graph or table can help.

Line 188: this is not academic and professional to use this type of writing in the paper: “Let’s see an example to illustrate how it works.”

The material and methods section needs to be rewrite and revised. For example 2.1.2 the paragraph starts without any introduction to the subject of this paragraph.  Also, it is a bit messy as the content in the methods section contains result and part of the discussion as well. (all needs to be revised).

The previous literature about modelling and their results needs to be added to the content and the result of this study needs to be compared with previous ones (also use more recent study) to clarify how your approach works better or able to solve the problem.

Conclusion: it is a big claim “we have confirmed (line 622)”.

Author Response

Thank you so much for your comments and suggestions. Please see the responses attached here.

Reviewer 2 Report

Dear Authors,

Thank you of the opportunity to review your manuscript entitled ‘Assessing Multifactorial Human-Nature Interactions Affecting the Environmental Quality through Models of Dynamic Networks: A System Thinking Approach’.

The topic of the paper is interesting, but a lot of provided information need more than just edit. However, as your idea is interesting I recommend you to work more on it and resend the paper again.

Please let me explain my main issues with the manuscript.

1. (Lack of) Structure of manuscript.

Except for the Introduction, all sections, like (2) Materials and Methods, (3) Results and Discussion, and (4) Conclusions, have to be rewritten.

A huge part of the (2) Materials and methods section belongs to (3) Results, as you present own results there, e.g. see Tables 1, 2, 3, 4, 5, or Figures 1 and 4. All this is supported by equations (e.g. line 192-196; 369-372). You themselves mention that at page 5 (line 187): 'The percentage increase is calculated by the following expression...'.

Furthermore, in this section (2) at a page number 13, Table 6 can be found. You entitled it as 'Environmental management actions to be chosen to reduce the current environmental trends'. How you developed recommendations and suggestions before results and discussion? Overall, this is definitely not a methodology but recommendations.

The (3) Results and Discussion section is mostly just repetition what I already knew from the previous section. For example, Table 7 is showing results from Table 2.

The Discussion part is completely missing. The purpose of the discussion is to describe the significance of your findings in light of what was already known about the research problem.

2. (Lack of) Logical consequence.

Your approach should follow the simple logic that the conclusion is the consequence of the premises. Now the reader is lost.

Figure 4 (page 8) is showing some numbers, but at this stage, I have no idea from where those numbers came from, and what they mean. The explanation can be found at the next page (9) at Table 3.

3. Title

While reading the title I thought that it will be a conceptual paper. But after I have gone through the manuscript, it seems to me more like a case study. The title should clearly explain the purpose (area) of the study.

4. Calculations.

I haven’t checked all calculations but in some cases, some mistakes can be found. For example, a percentage increase of LVC: line 194 and Table 8 (112.8%) and the same data from Table 2 (122.8%). Seems that 122.8% is the correct version (precisely: 122.8982), but according to mathematical rules (rounding), it should be 122.89%.

Similar, Table 3, Normalized Value for (delta)Pop-(delta)Waste is 0.662 and the same value in Table 4 is 0.663.

Alike, Normalized Value for Path_LVC (Table 4) is 0.635 and on line 517 is 0.634. (Please check angular value as well).   

5. Others

The line number of the comments refer to the line numbers added by the MDPI System.

54. CO2 – CO2; same line: 64, 75, 206, 214, 219, 222, 242, 370, 399, 414, 422, 458, 469, 471, 496, 504, 512, 534, 535, 536, 537, 544, 547, 548, 558, and Table 1, 2, 3, 4, 7,  Figure 7, 8.

129. It is: Sweeney and Sterman underlined the ability of systems thinking to represent and assess dynamic complexity [41].

It should be: Sweeney and Sterman [41] underlined the ability of systems thinking to represent and assess dynamic complexity.

153. the State of Morelos, Mexico.

185. Table 1.

Please explain acronyms, like CTR, LVC etc., e.g. under the table (as Note).

Please add one column, title it as a Reference and add a specific reference for the data.

205-209. What do you mean here?

226. Figure 1.

Those figures have to be larger, now most of them are no readable.

267. Figure 2.

How you choose those factors? What you mean by Environmental Education and how it can be placed at the same group as Population Growth? What about agriculture?

Furthermore, the other group Nature some components of the natural environment is missing. What about Soil, Land Relief, Wildlife, etc.?

368. Table 5.

Why those values are not equal? Why 20 and not 18?

370. Sometimes you are using Path_Waste and sometimes Path_WAS. Please keep one type of acronym.

401. Figure 6.

It is: Region at mid-risk [60 o-80 o]

It should be: Region at high risk [60o-80o)

It is: Region at mid-risk [80o-90 o]

It should be: Region at very high risk [80o-90 o]

420-423. This is repetition from line 397-400.

432. Figure 7 and 8 should be merged together. You might use dotted lines to explain the differences between an open and closed system.

508. ‘As we can see in Table 6, both the percentage ….’ We cannot see percentage at table 6.

With all of those concerns, I believe you have to put more work to make this submission appropriate for Sustainability.

Good luck!

Looking forward to seeing you revised manuscript. 

Author Response

(The authors gave the same response as above.)

Round 2

Reviewer 1 Report

The revised version  looks much better. However there is still few thinks to consider.

Abstract: the revised version looks much better but still it some sentences needs to be written clearly and more simple. Such as :

In this paper, we address the problem of understanding the multiple interactions between population increase and drivers and pressure factors to identify those interactions causing damage to human-nature interactions, which in turn affect the environment quality.

Line 19: needs to be more precise what type of effect: “ actions to reduce effects on the environment”. The abstract needs another revise, as the connection of sentences are missing or in some cases misleading.

Line 38: it is better to start with more general sentence first (as a background) and then start referencing to authors.

About the rest of manuscript the revisions and added sections improved the clarity of the manuscript, however the English writing and style needs to be revised (long sentences, misuse of punctuation marks, complex sentence structure). it is better to use more simple and shorter sentences to help reader follow the manuscript easily. 

Author Response

Dear Reviewer,

We really appreciate your interest in improving this work. So, thank you so much for your comments and suggestions. Please see the responses attached here.

Reviewer 2 Report

I carefully read the paper, and I have no further comments. Author done a good job.

Author Response

Dear Reviewer,

Thank you so much for your comments and suggestions.